# Rare Subtype of Endometrial Cancer: Undifferentiated/Dedifferentiated Endometrial Carcinoma, from Genetic Aspects to Clinical Practice

**DOI:** 10.3390/ijms23073794

**Published:** 2022-03-30

**Authors:** Hsiu-Jung Tung, Ren-Chin Wu, Chiao-Yun Lin, Chyong-Huey Lai

**Affiliations:** 1Department of Obstetrics and Gynecology, Chang Gung Memorial Hospital, Linkou Branch and Chang Gung University College of Medicine, Taoyuan 333, Taiwan; thjami@cgmh.org.tw (H.-J.T.); chiao.yun0101@gmail.com (C.-Y.L.); 2Gynecologic Cancer Research Center, Chang Gung Memorial Hospital, Linkou Branch, Taoyuan 333, Taiwan; renchin.wu@gmail.com; 3Department of Pathology, Chang Gung Memorial Hospital, Linkou Branch and Chang Gung University College of Medicine, Taoyuan 333, Taiwan

**Keywords:** undifferentiated, dedifferentiated, endometrial carcinoma, genetic expression, molecular target

## Abstract

Endometrial cancer (EC) is one of the most common gynecologic cancers worldwide. There were 417,367 newly diagnosed cases and 97,370 deaths due to this disease worldwide in 2020. The incidence rates have increased over time, especially in countries with rapid socioeconomic transitions, and EC has been the most prevalent gynecologic malignancy in Taiwan since 2012. The new EC molecular classifications of The Cancer Genome Atlas (TCGA) Research Network include clear-cell carcinoma, serous carcinoma, and carcinosarcoma, while undifferentiated/dedifferentiated EC (UDEC) is not mentioned, and most previous clinical trials for EC have not included UDEC. UDEC is rare, has an aggressive growth pattern, tends to be diagnosed at an advanced stage, and is resistant to conventional chemotherapy. In this review, case series or case reports on the clinical features and genomic/epigenetic and expression profiles on UDEC data are summarized in order to identify potential molecular targets for current and future research.

## 1. Introduction

Endometrial cancer (EC) is one of the most common gynecologic cancers worldwide. There were 61,880 newly diagnosed EC cases and 12,160 related deaths in the United States in 2019 [1], and EC has been the most prevalent gynecologic malignancy in Taiwan since 2012 [2]. The histologic subtype is prognostic; however, most previous studies on this topic have considered the high-risk subtypes to be serous, clear-cell, and carcinosarcoma. The Cancer Genome Atlas (TCGA) Research Network has provided new molecular classifications of EC [3], with profound impacts for clinical practice, but undifferentiated/dedifferentiated EC (UDEC) was not mentioned. Silva et al. reported that low-grade endometrioid carcinoma (grade 1 and 2) comprised 71% of EC, high-grade (HG), and non-endometrioid types in the remaining 29% [4]. Among the remaining 29% of the non-endometrioid type, the incidence was 13% serous carcinoma, 7% clear-cell carcinoma, and 9% undifferentiated EC (UEC) in the MD Anderson series [4]. The UEC is a solid-pattern tumor without specific morphologic evidence of epithelial differentiation [5]. It has an aggressive growth pattern that tends to be diagnosed at an advanced International Federation of Gynecology and Obstetrics (FIGO) stage and is resistant to conventional chemotherapy [4]. Dedifferentiated EC (DEC) is characterized by the coexistence of low-grade EC and UEC [6]. DEC has not been widely recognized due to its solid part usually being misdiagnosed as a grade 3 endometrioid EC [6]. A worse outcome than grade 3 EC was also found [6]. In a recent population-based study using the National Cancer Database of the United States (2004–2013), 1.1% of all ECs met the criteria of UDEC, which may reflect an underdiagnosis in earlier years [7].

UDEC harbors specific genetic features different from endometrioid carcinoma. Generally, endometrioid carcinoma is a hormone-dependent tumor that expresses hormone receptors that may respond to hormone therapy. Unfortunately, UDEC seldom has detectable hormone receptors and its tumorigenesis pathway has distinct features, such as microsatellite instability (MSI-H)/mismatch repair (MMR) protein [8,9] and the genomic inactivation of core components of the SWI/SNF chromatin-remodeling complex [10,11].

Thus, our aim is to review the literature concerning clinical features and molecular targets for novel therapy.

## 2. Histological Diagnosis, Immunohistochemistry, and Genetic Alterations

### 2.1. Histological Diagnosis, Immunohistochemistry

Histologically, UEC is characterized by a proliferation of dyscohesive tumor cells growing in patternless, sheet-like arrangements [12,13]. The tumor cells are typically medium-sized and monomorphic [12,13]. However, occasionally, marked nuclear pleomorphism, vague spindling, foci of abrupt keratinization, rhabdoid cells with abundant eosinophilic cytoplasm and eccentric nuclei, or a myxoid stroma can be noted [12,13]. A differentiated component, most commonly FIGO grade 1 or 2 endometrioid carcinoma, can be identified in approximately 40% of UEC cases and is termed “dedifferentiated carcinoma” [5]. The tumor histology poses a great diagnostic challenge and misdiagnosis is not uncommon, as it can mimic grade 3 EC, neuroendocrine carcinoma, carcinosarcoma, endometrial stromal sarcoma, undifferentiated sarcoma, melanoma, lymphoma, and plasmacytoma [4,5,13]. Immunohistochemically, UEC typically shows the focal expression of epithelial markers, especially CK18 and epithelial membrane antigen (EMA), the loss of cell adhesion molecules such as E-cadherin and claudin-4, and a lack of expression of Müllerian markers such as PAX8, ER, and PR [11,12,13,14]. A panel including PAX8, E-cadherin, cytokeratin, and EMA can be helpful to distinguish between UEC and grade 3 EC [5]. DNA MMR protein (MLH1, MSH2, MSH6, and PMS2) expression is lost in 44% of UEC cases [15]. Approximately 60% of UEC cases lose the expression of switch/sucrose non-fermenting (SWI/SNF) complex proteins, including ARID1B, SMARCA4 (BRG1), and SMARCB1 (INI1) [16,17]. Recent immunochemistry stain findings of the above markers are listed in Table 1. SWI/SNF complex deficiency ranged from 20.0 to 69.4%, and MMR deficiency (MMRd) ranged from 46.2 to 73.3% across different studies [8,11,18,19,20,21].

### 2.2. TCGA Molecular Classification in UDEC

TCGA documented four molecular categories of EC, known as *POLE*/ultramutated, microsatellite instability (MSI)/hypermutated, copy-number-low/*TP53*-wild-type, and copy-number-high/p53-abnormal [3]. A systemic review using pooled case series data reclassified into TCGA subgroups of UDEC (n = 73) found 44.0% of cases with MMR deficiency (MMR-d), 12.4% with *POLE* mutation, and 18.6% with p53 abnormality [22], for which the proportion of UDEC in the MSI group was lower than the previous report (about 60%) [23,24] under this calculation but was consistently higher than the overall EC (28%) in the TCGA cohort [3]. The proportion of UDEC in the *POLE* group (12.4%) was also higher than EC (7.3%) in the TCGA cohort without UDEC. As the majority of UDECs belonged to the MSI and *POLE* groups, a high mutational load is a feature of UDECs different from other high-risk histological subtypes of EC [22]. Since a high mutational load is a predictor of response to immunotherapy, UDECs may benefit from such treatment [8,25].

### 2.3. Switch/Sucrose Non-Fermentable (SWI/SNF) Complex Proteins

As component proteins of the switch/sucrose non-fermenting (SWI/SNF) chromatin remodeling complex, BRG1 (encoded by *SMARCA4*) and INI1 (encoded by *SMARCB1*) mutations are known to be related to several types of human malignancies [26,27]. They are key regulators of nucleosome positioning [28]. SWI/SNF complexes generate energy from ATP hydrolysis to slide or eject nucleosomes [29]. Inactivating mutations involving *SMARCA4* or *SMARCB1* cause the loss of BRG1 or INI1 expression, which is a significant subset of DEC [10,30]. The loss of BRG1 or INI1 is also associated with the loss of PAX8 and estrogen receptor (ER) in the undifferentiated component [23]. Mutations in the genes encoding SWI/SNF subunits often result in a loss of function. Missense mutations might be the most common type of mutation, for example, within the enzymatic ATPase domain of SMARCA4 [31]. The SMARCA4-deficient and SMARCB1-deficient DECs were reported to have a prevalence of approximately 50–68% [11,30]. Accompanied with these mutations, the majority (66–73%) of the undifferentiated component developed in a mismatch repair-deficient molecular context [11,30]. The SMARCA4-deficient and SMARCB1-deficient types were also associated with a worse outcome [11].

### 2.4. ARID1A and ARID1B

ARID1A and ARID1B are alternate but obligatory DNA-binding subunits of the SWI/SNF complex. Concurrent inactivating mutations that result in the loss of both ARID1A and ARID1B are expected to block chromatin remodeling functions, resulting in difficulty in binding and targeting DNA [23]. ARID1A and ARID1B co-inactivation appears to be an alternate mechanism to BRG1 or INI1, which results in the occurrence of the undifferentiated component [23]. Overall, ARID1A is the most frequently mutated SWI/SNF subunit across cancer types [29] and has a tumor-suppressive function, whereby it triggers cancer development by interfering with the DNA-damage response and cell cycle pathways [32,33]. Furthermore, several studies have also demonstrated that an ARID1A loss is associated with the activation of phosphatidylinositol-4,5-bisphosphate 3-kinase catalytic subunit alpha (PIK3CA) and the concurrent loss of PTEN expression, which both activate the PI3K/AKT/ mTOR cell cycle pathway [34,35,36,37]. A greater propensity for MMRd EC was observed in 75% of these ARID1A/ARID1B co-deficient DECs [23].

### 2.5. Deficiency of MMR Protein

Known MMR proteins include mutL protein homolog 1 (MLH1), postmeiotic segregation increased 2 (PMS2), mutS protein homolog 2 (MSH2), and mutS protein homolog 6 (MSH6). MMR defects were significantly associated with poor clinical outcomes in all types of EC. A higher tumor grade and the presence of lymphovascular space invasion (LVSI) were related to both epigenetic MMR defects and mutations [38]. However, MMRd was also related to better adjuvant treatment response and may have favorable survival [39]. Tafe et al. reviewed IHC in 17 cases of UEC from endometrium and ovary and showed that eight cases (47%) had MMRd UEC [13]. In a pooled cohort of UDEC (n = 73), 44% of cases were in the MMRd group [22].

### 2.6. Programmed Cell Death Ligand-1 (PD-L1)

Programmed death receptor (PD-1) and its ligand PD-L1 are known as important biomarkers for immune checkpoint inhibitors. As a co-inhibitory transmembrane receptor expressed by T cells, PD-1 can inhibit the proliferation, survival, and cytokine production of T cells by engaging with PD-L1 [40], resulting in immune escape. Extensive work has been undertaken to investigate PD-1 blockage in a variety of cancers, leading to clinical approval for treatment, for example, melanoma with pembrolizumab, and renal cell carcinoma with nivolumab [41]. However, in 2021, a large cohort with 833 samples reported that 45% positivity of PD-L1 (combined positive score, CPS > 1) may relate to a better outcome in high-risk EC (including seven UDECs) [42]. Some studies found a positive correlation with MMR-d- and PD-L1-positive tumors. Sloan et al. also found 100% PD-L1 positivity in MMRd patients, and 66% PD-L1 positivity in MMRp patients [43]. In the undifferentiated component, nine cases (53%) belonged to the MMRd group and all of these cases exhibited PD-L1 expression. Not only in all ECs, but also in UDEC, MMRd was significantly associated with PD-L1 expression (*p* = 0.026) [44]. However, in an analysis of 17 patients from a cancer center in Jordan, there was no correlation with MMR-p and PD-L1. However, PD-L1 expression in UDEC in this series accounted for 66.7% of cases (10/17 cases) [18], which suggests that DEC could be a good target for immune checkpoint inhibitors [44]. A DEC case was reported to have been successfully salvaged with pembrolizumab, with 15 months of progression-free survival [45].

### 2.7. Molecular Genetic Heterogeneity

The UDEC, as previously described, are genetically heterogeneous, with cases belonging to all four groups of TCGA classification. To further investigate genetic alteration, Rosa-Rosa, J.M. et al. carried out exome sequencing in 18 samples of UDEC. MSI hypermutation accounted for 44% (8/18) of UDEC cases, and *POLE* mutation for 11% (2/18); *PTEN* mutation accounted for 28% of endometrioid-like ECs (5/18); *TP53* mutation accounted for 11% of serous-like ECs (2/18); and there was one unclassified case [46]. The MSI group was larger than most series of sporadic EC (around 30%) [47]. One of the *POLE* groups in this series (two cases) harbored a *TP53* mutation. One-third of UEC in this series were low-copy-number endometrial carcinomas. The most frequent mutation was the *PTEN* mutation, associated with the PIK3CA pathway. Another report of sequencing analysis results also found UDEC with somatic mutations in *PIK3CA* (50%), *CTNNB1* (30%), *TP53* (30%), *FBXW7* (20%), and *PPP2R1A* (20%) [48].

## 3. Treatment and Prognosis

Due to the rarity of UDEC, there was little experience with novel treatment in case series. Real-world experience was limited to conventional chemotherapy and/or radiotherapy [49,50], as listed in Table 2 [11,21,45,49,50,51]. According to most clinical guidelines [52,53], comprehensive staging surgery should be performed in operable cases, including total hysterectomy with bilateral salpingo-oophorectomy, systematic lymphadenectomy, and infracolic omentectomy [53]. UDEC is classified as a type II EC, and warrants chemotherapy even at an early stage. Paclitaxel and carboplatin are suggested for stage IA with myometrial invasion every three weeks for 3–6 cycles. In patients with LVSI, brachytherapy may be arranged. For stage IB to IIIC2, pelvic radiotherapy with brachytherapy followed by chemotherapy is recommended. For positive pelvic or para-aortic lymph nodes, extended field radiotherapy should be arranged [19].

Despite such a heavy treatment setting, a fulminant recurrence pattern with short survival was reported by Han et al. All four patients in this series were LVSI positive. One stage II and one stage IIIB patient relapsed and died within a short interval. Although one patient (stage II) refused adjuvant therapy, the other patient with stage IIIB and vaginal wall extension received complete chemotherapy and radiotherapy, but still relapsed one month after she had completed treatment [50].

Pfaendler treated two chemoresistant UEC patients with a rapid progression pattern. As no hormone receptor was present, nor a microsatellite stable feature, case 1 with stage IA, LVSI-negative UEC, and 20% myometrial invasion, received six cycles of paclitaxel and carboplatin, but still had recurrence after 6 months, and was deceased 6 weeks after recurrence. In case 2, with MLH1 and PMS2 loss, pembrolizumab was tried for one dose, but the patient moved overseas [49].

Most cases of DEC coexisted with UEC of low-grade (FIGO grade 1-2) EC (DEC-LG). However, DEC can arise with a background of high-grade carcinoma (DEC-HG). A case series (n = 18) compared DEC-HG with DEC-LG, in which DEC-HG presented with advanced disease (stage III–IV) (7/11, 64%), whereas most cases of DEC-LG were stage 1 (6/7, 86%). A total of four DEC-HG patients recurred or died, while only one DEC-LG patient recurred, with a mean follow-up of 23.2 months [21]. Goh et al. reported seven DEC cases in Singapore, four of which had recurrent/progressive disease. A case with FIGO stage IIIA DEC had extensive recurrence affecting the thorax, abdomen, and pelvis after surgery and chemotherapy. She received pembrolizumab and gemcitabine and survived for a further 10 months under stable disease and the resolution of ascites. The other three cases of recurrent/progressive disease all died, even with salvage efforts. The only long-term survivor with a favorable outcome of an OS of 56 months was a stage IIIC1 patient. Her tumor size was relatively small (1.8cm), with a 25% UEC component and one (out of eight) lymph node metastasis of the endometrioid component. They also compared DEC patients with grade 3 endometrioid EC, which showed a 2-year survival OS of 31.3% in DEC versus 82.8% in their institution [45]. Whether the percentage of DEC and the grade of the DEC component in the primary tumor affects prognosis is still controversial.

The genomic inactivation of core SWI/SNF complex protein was recently shown to account for two-thirds of DEC/UECs. When comparing SWI/SNF-deficient tumors with SWI/SNF-intact tumors, a poorer prognosis in the SWI/SNF-deficient group was noted. The 2-year disease-specific survival (DSS) for stages I and II disease was 65 % in deficient tumors and 100% in intact tumors (*p* = 0.042). For stages III and IV disease, the medial survival was 4 months for SWI/SNF-deficient tumors (36 months for intact tumors, *p* = 0.0003). SWI/SNF-deficient tumors possessed a highly progressive disease pattern and were resistant to conventional chemotherapy [11]. Further targets or immune therapies should be attempted, aside from chemotherapy.

In a recent case series of 52 UDECs, in which MMR status was checked, 30 of 43 (69.8%) were MMRd. The 5-year disease-free survival was 80% for stage I/II, 29% for stage III, and 10% for stage IV. Multivariate analysis suggested that adjuvant chemotherapy, adjuvant radiotherapy, and a lower FIGO stage contributed to better disease-free survival (*p* < 0.05) [19].

## 4. Preclinical Data and Relevant Clinical Data

Several new drugs have been tested in UDEC in preclinical studies. Fibroblast Growth Factor Receptor (*FGFR2*) mutation was identified in approximately 10% of patients with primary endometrial cancer and more than 90% of patients with *PIK3CA* activation [54]. Thus, they tried a pan-FGFR inhibitor (BGJ398, infigratinib) with pan-PI3K inhibitors (GDC-0941, BKM 120) in endometrial cancer cell lines, which showed a significant increase in cell death and long-term survival when the PI3K inhibitor was combined with BGJ398 [54]. Lin et al. performed whole-exome sequencing from specimens of a uterine DEC patient, which revealed an *FGFR2* mutation and *CCNE2* amplification. The former was targeted with the FGFR inhibitor lenvatinib, while the latter was treated with the cell cycle inhibitor palbociclib in patient-derived xenograft (PDX) mice with UDEC. Compared with control mice, the tumor shrank significantly in the treatment arm [55]. The first approved FGFR2 inhibitor, pemigatinib, for treating cholangiocarcinoma [56] is now used in recurring *FGFR*-mutated solid tumors, including EC (NCT02393248, FIGHT-101). For a rare disease such as UDEC, the use of a precise genetic survey can find an effective target therapy for a specific molecular alteration of cancer.

Targeting the SWI/SNF complex may be the next step of treating UDEC, since two-thirds of UDEC patients had SWI/SNF deficiency [11,30]. Several molecules that inhibit SWI/SNF ATPase activity have been reported. The first published inhibitor was PFI-3 [57], but due to the lack of chemical stability in cellular systems over longer time periods, several other SMARCA2/4 BRD inhibitors are currently under development [58,59]. Proteolysis-targeting chimeras induce ubiquitin transfer onto target proteins, thereby marking them for proteasomal degradation. This was applied to target BRD9, a SWI/SNF complex subunit [33]. For example, AU-15330 induced a potent inhibitor of tumor growth in xenograft models of prostate cancer, and synergized with the androgen receptor antagonist enzalutamide, even inducing disease remission in castration-resistant prostate cancer models without toxicity [60].

Aurora A was reported to be a potential therapeutic target in ARID1A-deficient colorectal cancer cells [61]. Wild-type ARID1A downregulated the expression of Aurora A, while ARID1A loss led to increased expression levels of Aurora A and activated the cell division cycle. Thus, the inhibition of Aurora A with a kinase inhibitor induced a G2/M phase arrest, followed by apoptosis [33]. Alisertib as an Aurora A kinase inhibitor has recently been used in phase 3 randomized trials for hematologic malignancy. While it showed disappointing early results, it has been investigated in a number of cancer types, as a monotherapy or in combination [62]. In a randomized phase 3 trial, alpelisib plus fulvestrant were tested in *PIK3CA*-mutated, hormone-receptor-positive advanced breast cancer, which prolonged progression-free survival (11.0 months vs. 5.7 months) for longer than in the placebo plus fulvestrant arm [63].

The enhancer of zeste homologue 2 (EZH2) inhibitor tazemetostat is under testing in adults with diffuse large B cell lymphoma or SMARCB1-negative or SMARCA4-negative solid tumors, and in pediatric patients with rhabdoid tumors, synovial sarcoma, epithelioid sarcoma, or other cancers with *SMARCB1*, *SMARCA4*, or *EZH2* mutations [64,65,66,67]. A trial was performed in patients with epithelioid sarcomas (nearly all tumors have homozygous deletion of *SMARCB1*), which demonstrated a response rate of 15%, and of the patients who responded, 67% had a response lasting 6 months or longer [68]. However, applications in EC with the SWI/SNF mutation remained unknown, until now.

Finally, as we know, MMR-d and PD-L1 account for about 60% of these UDECs. Immune checkpoint inhibitors should be considered, although there is a lack of trials or large databases grounded in real world experience specific for UEC/DEC. In recent EC trials, KENOTE-158 used pembrolizumab monotherapy for noncolorectal high-microsatellite-instability/MMR-d cancers, which included 49 cases (21.0%) of endometrial cancer and yielded the best response rate (57.1%) and longest progression-free survival (25.7 months) among gastric, cholangiocarcinoma, pancreatic, small intestine, ovarian, and brain cancer [69]. Pembrolizumab plus levatinib showed a promising antitumor effect in KEYNOTE-146, which included one UDEC patient [70], and has further been verified by a phase 3 trial, KEYNOTE-775, presented at the 2021 SGO conference [71]. Another anti-PD-1 antibody, dostarlimab, showed a 42% objective response rate in MMRd EC in its phase 1 trial [72]. APR-246, a prodrug that binds to cysteine residues in mutant *p53* and restores its wild-type function, showed synergistic effects with chemotherapy in ovarian cancer cell lines and resensitized platinum-resistant ovarian cancer cells [73]. Adavosertib (MK-1775), a potent, small-molecule WEE1 kinase inhibitor, showed an antitumor effect in combination with chemotherapy and radiotherapy in preclinical studies. There are several ongoing trials of APR-246 and adavosertib in ovarian cancer [73]. Since a *TP53* mutation is one of the features of UEDC [22,48], agents targeting the mutant p53 pathway can also be explored.

We summarize the potential therapeutic agents targeting members of these pathways in Figure 1. Target/immune therapies are collected and depicted in Table 3.

## 5. Conclusions and Future Directions

UDECs are clinically aggressive diseases with chemoresistance. Recurrent disease almost always has a dismal outcome. The prevalence of UDEC might have been underestimated, and the prevalence of DEC-HG versus DEC-LG needs more research for clarification. UDEC patients should be encouraged to enroll in EC clinical trials, and the sponsors should be able to conduct subgroup analyses for UDEC. Since about half of these tumors have MMR-d features and the expression of PD-L1, immune therapy may be eligible for treatment options. The SWI/SNF complex plays a crucial role, especially in patients with a poor prognosis. Targeting the SWI/SNF pathway and subunit may open a new therapeutic window for UDECs. Comprehensive genetic sequencing is also helpful in finding possible potent drugs.

## 6. Materials and Methods

We searched articles published up to February 2022 in the PubMed (https://pubmed.ncbi.nlm.nih.gov), UpToDate, and Google Scholar databases by using the following keywords: undifferentiated or dedifferentiated carcinoma of uterus, genetic analysis, molecular markers, and prognosis and treatment of UDEC. The most relevant reviews and case series were included, and the reference lists of these selected articles were read. By focusing on future directions, the latest case reports or preclinical data of special or target therapies were also presented.

## Figures and Tables

**Figure 1 ijms-23-03794-f001:**
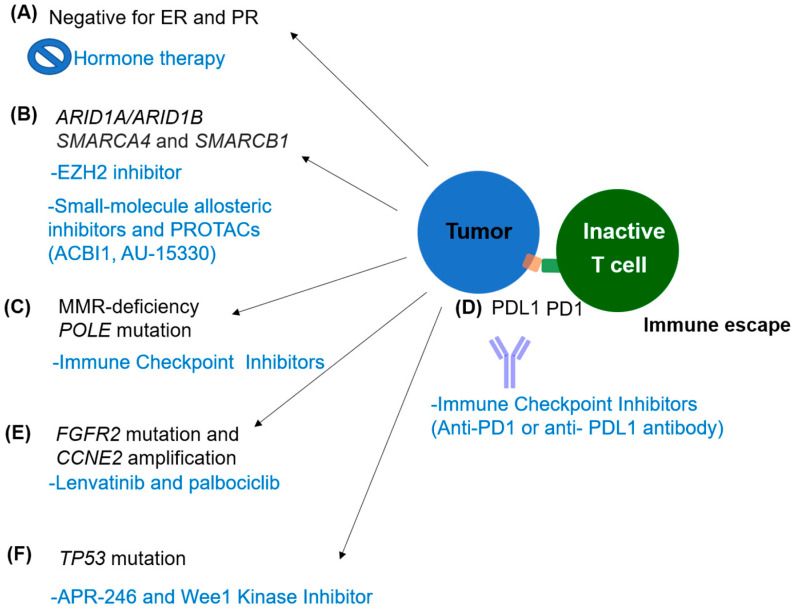
Immunologic and genomic hallmarks of DEC. (**A**) DEC is not amenable to hormone therapy owing to its loss of ER and PR [5,6]. (**B**) Mutations in genes encoding subunits of the SWI/SNF chromatin remodeling complex could be targeted by small-molecule allosteric inhibitors and proteolysis-targeting chimeras (PROTACs) [29,60]. (**C**,**D**) Defects in MMR and the increased expression of PDL1 could be a target for immune checkpoint inhibitors [8,29,69,72]. (**E**) Mutations in *FGFR2* and the amplification of *CCNE2* could be a target for lenvatinib and Palbociclib [55]. (**F**) Mutation in *TP53* could be a target for APE-246 and Wee1 kinase inhibitor [73]. For each inhibitor against the gene, the rationale of cancer treatment is described (signified by blue words).

**Table 1 ijms-23-03794-t001:** Clinicopathologic features of UDEC.

Author	Case Number/UEC or DEC	SWI/SNF Deficiency (%)		ARID1A/1B Co-Deficient	MMR-d/-p (%)	*POLE* Mutated/WT	PDL1 +/− (%)	p53 +/−	Note
Hacking et al., 2019 [8]	14/ 4 UEC 10 DEC	NA		NA	8/6 (57.1%)	NA	7/7 (50.0%)	NA	PDL1 not expressed in MMR-p tumors
Tessier-Cloutier et al., 2021 [11]	82/ 53 DEC 29 UEC	56 SWI/SNF deficiency (68.3%)		25 (44.6%)	38/18 (67.9%)	2/54 (3.6%)	NA	NA	
		26 SWI/SNF intact (31.7%)		0 (0)	12/14 (46.2%)	4/22 (15.4%)	NA	NA	
Al-Hussaini et al., 2018 [18]	17/ 8 UEC- pure 4 UEC-mixed 5 DEC	BRG1/SMARCA4 loss: 3 ^a^ (20.0%)		NA	11/6 (64.7%)	NA	10/7 (66.7%)	8/7 ^a^ (53.3%)	^a^ Two patients not applicable
Hamilton et al., 2022 [19]	52/ 17 UEC 35 DEC	34/15 ^b^ SWI/SNF deficiency (69.4%)		22/27 ^b^ ARID1A/1B co-deficient (44.9%)	30/13 ^c^ (69.8%)	NA	NA	NA	^b^ 49 patients checked ^c^ 43 patients checked
Hoang et al., 2016 [20]	35 DEC	20 Loss of BRG1 or INI1 (57.1%)		NA	13/7 (65.0%)	NA	NA	1/19 (5.0%)	
		15 BRG1/INI1 intact (42.9%)		NA	11/4 (73.3%)	NA	NA	3/12 (20.0%)	
Busca et al., 2020 [21]	18 DEC/ 11DEC-HG 7DEC-LG	11 DEC-HG	3/6 (33.3%) (BRG1 loss/intact)	NA	6/5 (54.5%)	NA	NA	5/6 (45.4%)	
		7 DEC-LG	1/4(20.0%) (BRG1 loss/intact)		4/2 (66.6%)			0/5 (0)	

Abbreviations: MMR-d, mismatch repair-deficient; MMR-p, mismatch repair-proficient; NA, not applicable. ^a^ although total patients is 17, the ratio was counted with 3/15 due to two patient’s data not applicable. ^b^ total patients is 52, the ratio was counted with 34/49 due to 49 patients’ data available. ^c^ total patients were 52, but there were 43 patients’ data available thus ratio counted with 30/43.

**Table 2 ijms-23-03794-t002:** Reported treatment in case series and clinical outcomes in UDEC.

Author	Case Number	Diagnosis Age (Mean, y-o)	Stage (Cases)	Treatment (Cases)	DFS (Months)	OS (Months)	Outcome
Tessier-Cloutier et al., 2021 [11]	82	61	SWI/SNF-deficient I (22) II (3) III (15) IV (16)	NA	NA	2-year DSS for stage I–II: 65% Stage III–IV: 3%	DOD (38) NED (18)
		64	SWI/SNF-intact I (13) II (2) III (7) IV (4)	NA	NA	2-year DSS for stage I–II: 100% Stage III–IV: 61%	DOD (10) AWD (1) DOOC (2) NED (13)
Busca et al., 2020 [21].	18	68	DEC-HG I–II (4) III–IV (7)	Surgery (11) CT (7) RT (9)	1.7 2.1 10.7 NA	NA NA 37.7 21.7	REC (2) DOD (2) NED (7)
		72	DEC-LG I–II (6) III–IV (1)	Surgery (7) CT (3) RT (7)	6.3 (recurrent patient)	17+ (recurrent patient)	REC (1) NED (6)
Goh et al., 2020 [45]	7	56	II (1) III (5) IV (1)	Surgery + TP × 6+RT Surgery + TP or TC × 5 – 6 NACT	15 58, 9, 5, 1, 2 0	15+ 58+, 25, 21+, 6, NA 9	DOD (3) AWD (1) NED (2) NA (1) *
Pfaendler et al., 2019 [49]	2	56	I (1) III (1)	Surgery + TP × 6 Surgery + TP × 3	6 4	7 NA	DOD (1) NA (1) *
Han et al., 2017 [50]	4	61	IA (1) II (2) IIIB (1)	None (1) Pt refuse (1), RT (1) CT + RT (1)	NA 1, NA 1	19+ 7 weeks, 39+ 10	DOD (2) NED (2)
Silva et al., 2006 [51]	25	51 (median)	I (14) II (1) III (6) IV (4)	Surgery (24) CT (18) RT (4)	NA	7 6–8+	DOD (15) AWD (6) NA (3)

Abbreviations: TP, paclitaxel + carboplatin; TC, paclitaxel + cisplatin; DFS, disease-free survival; OS, overall survival; RT, radiotherapy; CT, chemotherapy; NA, not applicable; REC, recurrence; DOD, died of disease; NED, no evidence of disease; AWD, alive with disease; DOOC, died of other cause; DSS, disease-specific survival; y-o, year-old; * move to other country/loss to follow up.

**Table 3 ijms-23-03794-t003:** Potential targets and existing inhibitors or activators in UDEC.

Mechanism	Subunit/Genetic Target	Medication	Reference
Mismatch repair deficiency PDL1/PD1 pathway			
	Anti-PD1 antibody	Pembrolizumab	[69]
	Anti-PD1 antibody	Dostarlimab	[72]
Anti-FGFR2		BGJ398, infigratinib.	[54]
ARID1A	Aurora A	Alisertib	[33]
SWI/SNF-polycomb antagonism	PCR2, EZH2	Tazemetostat	[33]
SWI/SNF deficiency	Synthetic lethal interaction	Inhibitors	[33]
SMARCA4	CDK4/6	Palbociclib, abemaciclib, ribociclib	
	Aurora A	Tozasertib, alisertib	
ARID1A	PARP	Talazoparib, olaparib, rucaparib, veliparib	
	Abl, Src, c-KIT	Dasatinib	
SMARCB1	HDAC	Panobinostat	
	UBE2C	Ixazomib, bortezomib	
*TP53* mutation			
	Mutant p53 cysteine residue	APR246	[73]
	Wee1	Adavosertib	[73]

## Data Availability

Not applicable.

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
