# Peer review of "Rare Subtype of Endometrial Cancer: Undifferentiated/Dedifferentiated Endometrial Carcinoma, from Genetic Aspects to Clinical Practice"

_ijms, 2022, doi:10.3390/ijms23073794_

Round 1

Reviewer 1 Report

This is an interesting, well written, review article about Undifferentiated/Dedifferentiated Endometrial Carcinoma, representing an accurate analysis of genetic characteristics of this rare type of EC and possible molecular targets for the future.

I have the following comments to the Authors:

  • Authors should check for some mistakes in the text:
  • Line 18: “malignancy” instead of “malignancies”
  • Line 31: “were” should replace “was”;
  • Line 32: “United States”
  • Authors should avoid using “but” at the beginning of a period, using “however” instead (lines 72 and 153);
  • Authors should check for syntax in lines 120-121, since period is not very clear as written;
  • Authors should check for syntax in lines 144-145, since period is not very clear as written;
  • Line 156: there is a repetition (“15 months”);
  • Line 159: “investigate” instead of “investigated”;
  • Authors should check for syntax in lines 160-162, since period is not very clear as written
  • Line 174: word “salpingoophorectomy” is written wrong;
  • Line 192: word “chemoresistance” is written wrong
  • Authors should check for syntax in lines 237-239, since period is not very clear as written;
  • Line 245: “inhibit” instead of “inhibiting”
  • Line 266: “is” instead of “it”
  • Line 270: “has” instead of “is”
  • Since the article is a review of the literature, in order to make the research reproducible, Authors should describe the search strategy, the process of study selection, andhow was assessed the risk of bias within studies.

Reviewer 2 Report

The manuscript entitled “Rare Subtype of Endometrial Cancer: Undifferentiated/dedifferentiated Endometrial Carcinoma, from Genetic Aspect to Clinical Practice” reviewed the Undifferentiated/dedifferentiated Endometrial Carcinoma. This manuscript has some issues that need to be addressed:

  • The English should be polished since some sentences are hard to read and understand.
  • Genes should be italicized.
  • The number of decimal cases should be the same within tables and sentences (e.g. lines 78 to 80; lines 87 to 91; Table 1).
